# Atypia and Follicular Lesions of Undetermined Significance in Subsequent Biopsy Result: What Clinicians Need to Know

**DOI:** 10.3390/jcm10143082

**Published:** 2021-07-12

**Authors:** Krzysztof Kaliszewski, Dorota Diakowska, Marta Rzeszutko, Łukasz Nowak, Urszula Tokarczyk, Jerzy Rudnicki

**Affiliations:** 1Department of General, Minimally Invasive and Endocrine Surgery, Wroclaw Medical University, Borowska Street 213, 50-556 Wroclaw, Poland; urszula.tokarczyk@student.umed.wroc.pl (U.T.); jerzy.rudnicki@umed.wroc.pl (J.R.); 2Department of Nervous System Diseases, Faculty of Health Science, Wroclaw Medical University, Bartel Street 5, 51-618 Wroclaw, Poland; dorota.diakowska@umed.wroc.pl; 3Department of Pathomorphology, Wroclaw Medical University, Marcinkowski Street 1, 50-368 Wroclaw, Poland; rzemarta@wp.pl; 4Department of Urology and Urological Oncology, Wroclaw Medical University, Borowska Street 213, 50-556 Wroclaw, Poland; lllukasz.nowak@gmail.com

**Keywords:** atypia of undetermined significance, follicular lesion of undetermined significance, thyroid nodules, risk of malignancy, surgery

## Abstract

Atypia and follicular lesions of undetermined significance (AUS/FLUS) is the most controversial category of The Bethesda System. The risk of malignancy (ROM) in this group is estimated as 5–15%, however, the occurrence of two or more subsequent biopsy results with AUS/FLUS diagnosis makes these clinical situations more complex. We evaluated the ROM and prognostic value of aggressive ultrasound (US) features in 342 patients with thyroid nodules (TNs) with subsequent biopsy results of AUS/FLUS. We assessed US features and compared them with the final histopathological diagnosis. Overall, 121 (35.4%) individuals after first AUS/FLUS diagnosis underwent surgery and 221 (64.6%) patients had repeated biopsies. The ROM after first, second, and third biopsies with subsequent AUS/FLUS diagnosis were 7.4%, 18.5%, and 38.4% respectively. We demonstrated significantly higher rates of occurrence of aggressive US features in patients with malignancy (*p* < 0.0001). The age <55 years old was also a significant risk factor for TC (*p* = 0.044). Significant associations were found between aggressive US features and malignancy in patients after first diagnosis of AUS/FLUS (*p* < 0.05). The juxtaposition of US features with the number of biopsy repetitions of TN with consecutive AUS/FLUS diagnoses may simplify the decision-making process in surgical management. Two or three consecutive biopsy results with AUS/FLUS diagnosis increases the ROM.

## 1. Introduction

Ultrasound-guided fine-needle aspiration biopsy (UG-FNAB) is the most common and practical diagnostic tool for thyroid nodules (TNs) management [1]. Its high value as the preoperative TNs evaluation is reported because it gives clinicians the most reliable information concerning the potential malignant nature of the thyroid lesions. According to American Thyroid Association (ATA) guidelines [2], UG-FNAB is the most accurate and cost-effective method for TNs evaluation.

To enhance the communication and understanding between pathologists and clinicians for TNs treatment, in 2009, The Bethesda System for Reporting Thyroid Cytopathology (TBSRTC) was introduced [3]. After its modification in 2017, TBSRTC has been widely adopted worldwide [4,5]. The criteria and recommendations included in TBSRTC are generally straightforward. However, in unique cases, challenges might occur. This is due to the presence of intermediate results, which are contained in this classification [3,4,5]. Many authors confirm that the intermediate category is a major limitation of classifying UG-FNAB in assessing TNs [1]. These intermediate diagnoses of TBSRTC are atypia of undetermined significance (AUS) and follicular lesion of undetermined significance (FLUS), forming together category III [3,4]. AUS and FLUS include a heterogeneous group of various subcategories characterized by nuclear and architectural atypia [6].

The overall malignancy rate in the third category of TBSRTC, according to some authors, is much higher than reported initially [7,8]. What is more, it may vary between some concrete clinical scenarios [6]. Thus, AUS and FLUS do not provide a differential diagnosis between malignant and benign lesions. Some authors admit that it is the most controversial category of this classification [9]. The occurrence of two or more subsequent biopsy results with AUS/FLUS diagnosis makes these clinical situations even more complex. Surgical management of the AUS/FLUS category seems to vary and depends on the experience of the clinicians. Thus, currently, category III of TBSRTC represents a considerable challenge in the field of TNs treatment. It is difficult to predict the actual malignancy rate of TNs with AUS/FLUS diagnosis unless patients were qualified for therapeutic/diagnostic surgery and histopathology. Although it is recommended to consider some clinical features in the decision-making of AUS/FLUS management to avoid unnecessary surgery and not to overlook malignancy, specific ultrasound (US) characteristics of TNs with subsequent AUS/FLUS diagnoses are not entirely defined.

We performed a retrospective analysis of the individuals with TNs subsequently assigned to category III of TBSRTC and estimated if some aggressive US features may be taken under consideration before a therapeutic decision. We assessed how to properly select the individuals who should undergo diagnostic/therapeutic surgery. We assessed ultrasound features, which may help clinicians to make proper decision in AUS/FLUS TNs management cases.

## 2. Materials and Methods

### 2.1. Study Population

We retrospectively analyzed 5024 medical records of patients admitted and surgically treated in the 1st Department and Clinic of General, Gastrointestinal and Endocrine Surgery at the Wroclaw Medical University in Poland due to TNs between January 2008 and December 2018.

The study protocol was approved by the Institutional Review Board and Ethics Committee of Wroclaw Medical University, Wroclaw, Poland (No: KB 783/2017). All of our patients provided admission informed consent, which stipulated that the results may be used for research purposes. The data were analyzed retrospectively and anonymously from established medical records. The authors did not have access to identifying patient information or direct access to the study participants.

The steps for patient selection are presented in Figure 1. From the initial group of patients (*n* = 5028), we extracted data from 342 (6.8%) patients with AUS/FLUS diagnosis. In each of the medical records of these individuals, we evaluated the numbers and diagnosis of all biopsies performed during the observational study time. All patients were qualified to UG-FNAB in accordance with American Thyroid Association Management Guidelines for Adults Patients with Thyroid Nodules and Differentiated Thyroid Cancer [2]. All US examinations were performed by two radiologists with a minimum of 10 years of experience in thyroid ultrasonography. All US features of all TNs of every single patient were accurately described and introduced into the medical data base formed for this study. We evaluated five ultrasound (US) features of TNs with AUS/FLUS diagnosis (microcalcifications, hypoechogenicity, irregular margins, taller-than-wide shape, and high vascularity defined as intranodular flow with multiple vascular poles chaotically arranged) and compared them with the final histopathological diagnosis.

### 2.2. Statistical Analysis

Statistical analyzes were performed using Statistica v.13.3 (Tibco Software Inc., Palo Alto, CA, USA). Descriptive data were presented as the number of observations and percent mean and standard deviation (±SD) or median and IQR (interquartile range). The distribution of quantitative data was analyzed by Kolmogorov–Smirnov tests. In comparing characteristics of patients with benign and malignant tumor nodules, Fisher’s exact test was used. Multiple logistic regression analyses with the Wald test were conducted to identify risk factors predicting the occurrence of malignant thyroid tumors. Analyses of risk factors in small groups of patients were performed by calculating odds ratios and 95% confidence intervals (±95% CI) and were shown as forest plot graphs. A two-tailed *p*-value of <0.05 was considered to be statistically significant.

## 3. Results

### 3.1. Patients’ Characteristic

Clinical characteristics of the 342 patients with AUS/FLUS diagnosis are presented in Table 1.

There are 284 (83.0%) females and 58 (17%) males with a mean age of 51 ± 15 years old. Among them, 121 (35.4%) patients (Group A) underwent thyroidectomy after the first AUS/FLUS diagnosis, and 221 (64.6%) individuals (Group B) were qualified to repeat UG-FNAB (Figure 1). In group A, 37 (30.6%) individuals presented a minimum of four aggressive US features, and nine (7.4%) of them had a malignant tumor (Figure 1). The risk of malignancy (ROM) was 7.4%. In group B, after the second biopsy, 40 patients had an AUS/FLUS diagnosis. Twenty-seven (67.5%) of them underwent surgery, but 13 (32.5%) were qualified for a third biopsy. In the group of 27 patients operated after the second AUS/FLUS diagnosis, 10 (37.0%) had a minimum of three aggressive US features, and five (18.5%) of them were diagnosed as malignant. ROM was 18.5%. In the group of patients qualified for the next (third) biopsy, 13 had AUS/FLUS diagnosis, and they all underwent operations. Among them, nine (69.2%) TNs presented a minimum of two aggressive US features, and five (38.5%) of them were finally diagnosed as malignant. ROM in this group was 38.4%.

Of 221 total patients, 181 (81.9%) qualified for a repeated biopsy after the first procedure changed the AUS/FLUS diagnosis to II (*n* = 98), IV (*n* = 61), and V (*n* = 22) Bethesda categories. Analysis of US data confirmed higher rates (2:1, 3:1) of the absence of negative US features in a group of 342 patients (Table 1).

### 3.2. Association between Age, Sex, and US Features, and Occurrence of TN Malignancy

The analyses of demographics and the US features as risk factors of TN malignancy for all patients (*n* = 161) with a final diagnosis of AUS/FLUS are presented in Table 2.

Multiple logistic regression analyses demonstrated significantly higher rates of microcalcifications, hypoechogenicity, irregular margins, and taller-than-wide shape in patients with TN malignancy than in patients with benign disease (*p* < 0.0001 for all). In addition, the age below 55 years old was also a significant risk factor for cancer presence (*p* = 0.044). The probability of nodule size < 2 cm was 100% for patients with TN malignancy and logistic regression analysis for this variable was not performed.

We compared the demographic and US risk factors between histologically confirmed benign and cancer subgroups of AUS/FLUS patients after the first, second, and third biopsies. As shown in Figure 2, significant associations were found between microcalcifications, hypoechogenicity, high vascularity, irregular margins, taller-than-wide shape, and TN malignancy in the patients after the first diagnosis of AUS/FLUS (*p* < 0.05 for all).

After the second biopsy, the thyroidectomy group showed higher malignancy rates in hypoechogenicity and taller-than-wide shape (*p* < 0.05) (Figure 3).

Analysis of the relationship between risk factors and cancer presence in the patients after the third biopsy demonstrated a lack of statistically significant results, although OR data are promised (Figure 4).

We could not assess US features in presurgical TN diagnosis in these patients because the subgroup size was too small.

## 4. Discussion

ROM of AUS/FLUS diagnosis according to TBSRTC (2017) is estimated to be 10–30% if noninvasive follicular thyroid neoplasm with papillary-like nuclear features (NIFTP) is considered malignant, and 6–18% if excluding NIFTP from the malignant group [3,4]. However, some authors suggest higher malignancy risk, reaching up to 50% [10], and even after excluding NIFTP [11]. Based on the various ROM rates, the clinical practice towards AUS/FLUS category varies. Such results produce a unique clinical dilemma concerning AUS/FLUS TNs evaluation.

Although many clinicians manage patients with category III of TBSRTC, surgeons must decide on radical treatment or further observation. Based on our own experience, we performed the retrospective analysis to identify ultrasound and clinical parameters for risk stratification to help in the decision-making process of either radical treatment or a further clinical follow-up. Many authors have confirmed the difficulty of managing AUS/FLUS TNs, as the thyroid lesions lie between the extremes of benignity and malignancy [8]. They propose management options for nodules such as observation, surgery, or repeat UG-FNAB. Marin et al. [9] suggest that repeat UG-FNAB in cases of AUS/FLUS category increases detection of follicular adenomas. Focal or extensive but mild cytological and architectural atypia are insufficient to be assigned to a higher category, and atypia in follicular/lymphoid cells is classified as category III of TBSRTC [3]. Consequently, most neoplastic and non-neoplastic TNs can present AUS/FLUS features. However, high-grade tumors are rarely included in this category [12,13]. In this category, some individual or regional differences in the interpretation of AUS/FLUS criteria may be observed [13,14]. In our study group, the frequency of category III was 6.8% against the recommended 7% cut-off [3]. However, we analyzed only the individuals with AUS/FLUS category who underwent surgical resection.

The other differences regarding individual or regional AUS/FLUS interpretation may be observed in its management classified to this category. Guleria et al. [14] revealed several differences in AUS/FLUS TNs’ clinical practice in analyzed geographical regions. Although the total incidence of Bethesda category III was similar in the investigated countries, the combined resection rate and ROM of AUS/FLUS TNs varied. The authors noticed that TNs assigned to category III of TBSRTC were resected more often in India and Western countries than in Asia [14]. They also saw higher ROM in the analyzed areas when compared to recommended values [4,5]. Our analysis demonstrated that the group of patients with one AUS/FLUS diagnosis ROM was 7.4% and dramatically increased after the second and third AUS/FLUS diagnosis. They were estimated at 18.5% and 38.4%, respectively. In our study, we estimated that all TNs with AUS/FLUS diagnosis and malignancy revealed after histopathology examination were below 2 cm in size. This was nine patients after the first UG-FNAB and 10 patients after the second or third biopsy. Such results might be caused by the rather small number of analyzed patients with AUS/FLUS diagnosis, which finally were recognized as malignant tumors. The majority of them harbored US features connected with the higher risk of malignancy, so they were resected at the lower stage of disease.

Clinicians in various countries may take a less or more selective surgical approach in cases of TNs with indeterminate cytology. Some avoid the overtreatment phenomenon and prefer conservative management, such as active follow-up of AUS/FLUS individuals [15]. Such attitudes are primarily observed in some Asian countries. Guleria et al. [14] noticed that in the West and India, the resection rate of AUS/FLUS TNs is relatively high (42% and 53%, respectively). They added that such patients undergo surgery because of three main reasons. Firstly, repeated UG-FNAB often leads to changing the category to the higher one due to the patient’s choice and institutional experience. In our analysis, 181 (52.9%) patients with previously estimated AUS/FLUS diagnosis changed the Bethesda category to another after the second UG-FNAB. A total of 83 (24.2%) individuals changed the III category for the higher (IV or V).

There is always a concern regarding the loss of the individuals qualified for active follow-up for developing countries [14]. Because of the controversial approach to AUS/FLUS diagnosis, clinical features and molecular tests have been recommended in the decision-making process. However, repeated UG-FNAB is also suggested [2,4]. Zhou et al. [16] revealed that UG-FNAB combined with *BRAF600E* mutation might significantly improve the detection rate of malignancy in TNs assigned preoperatively to the III category TBSRTC. They confirmed that in 57% of AUS/FLUS TNs, the histopathological diagnosis was papillary thyroid cancer (PTC). This might be valuable preoperative clinical information. However, genetic tests are not routinely performed in many AUS/FLUS patients. In our study, the analyzed patients did not have molecular tests performed, so we did not assess any correlation.

It was estimated that patients with AUS/FLUS TNs might also be referred to surgery instead of repeat UG-FNAB based on clinical and ultrasound findings [3]. According to American Thyroid Association (ATA) 2015 guidelines [2], patients with AUS/FLUS TN diagnosis should undergo a repeat UG-FNAB or molecular test to enhance ROM rather than be sent to diagnostic surgery [2]. However, if molecular tests or repeat UG-FNAB cannot be performed, active follow-up or surgical treatment can be adopted [2]. The decision on surgery should be made based on clinical risk factors, ultrasound features, or patient preference.

Some authors estimated that two consecutive UG-FNAB with AUS/FLUS diagnoses increases ROM at least 30% than a single biopsy with one Bethesda III diagnosis [17,18,19]. Kaya et al. [20] estimated AUS/FLUS diagnosis in 17.4% of all biopsied patients. However, in the individuals who had repeated UG-FNAB and got a second consecutive AUS/FLUS diagnosis, the malignancy rate was 27.5%. In our study, we observed and confirmed that repeated biopsy with consecutive AUS/FLUS diagnoses increased ROM from 24.3% after the first UG-FNAB to even 55.5% after the third procedure. On the other hand, some authors state that there are no statistical differences in the malignancy rate of TNs after one or two UG-FNAB with one or two AUS/FLUS diagnoses [17,21,22]. They add that repeated biopsy or surgical treatment should always be performed under clinicians’ decisions. In our study, we partly confirmed this observation. Our results are also in concordance with other authors’ analysis, who show that irregular margins, hypoechogenicity, microcalcifications, taller-than-wide shape, high vascularity, nodule size, and age of the patients at the time of AUS/FLUS diagnosis might be helpful in malignancy prediction [23,24]. Other researchers noticed that malignant nodules which had AUS/FLUS diagnosis before surgery had significantly larger sizes than benign ones, so they recommended surgical treatment rather than repeat UG-FNAB in the larger TN with AUS/FLUS category [22]. Kuru et al. [25] recommended surgical treatment in AUS/FLUS TNs with solid structure, hypoechogenicity, irregular margins, and microcalcifications. The authors estimated the higher ROM in the presence of these ultrasound characteristics combined with AUS/FLUS diagnosis. Hong et al. [26] presented a study in which they evaluated AUS/FLUS TNs with intermediate and high ultrasound characters (stage 4 or 5) according to the Korean Thyroid Imaging Reporting, and Data System (KTIDS) had increased ROM in the range of 30–90%. Based on these results, they recommend surgical treatment than repeat UG-FNAB or observation. Kaya et al. [20] suggested that in such cases, the recommended surgical treatment should be a lobectomy. However, they also state that this approach can be changed according to clinical risk factors, ultrasound characteristics, molecular test results, or patient preference. Indeed, until recently, TNs with AUS/FLUS diagnosis were commonly qualified to repeat UG-FNAB or diagnostic surgery. However, three-quarters of them turned out to be benign on histopathology indicating unnecessary thyroidectomy [27]. Scientific progress in the genetics of TC genesis has led to the development of several molecular tests to improve and complement cytology, which enhanced the risk-based stratification of indeterminate biopsy results [28]. Many commercial tests use mRNA expression of UG-FNAB samples while others detect DNA alterations [29]. Some authors state that the next generation sequencing development and inclusion of the other genetic markers such miRNA may improve the diagnostic accuracy of molecular tests [29].

Correlation of ultrasound characteristics and UG-FNAB results provide a comprehensive evaluation of TNs. What is more, US examination facilities to characterize the US features associated with an increased risk of TNs malignancy. It is especially valuable in cases of undetermined biopsy results. Al-Salam et al. estimated that UG-FNAB and ultrasonography are key tools in predicting malignancy in TNs [30]. The authors suggest that because TNs with Bethesda III and IV diagnoses may present a higher risk of malignancy, greater US vigilance is required [30].

In summary, recognizing the ultrasound and clinical features is critical in developing an appropriate, tailored management approach to AUS/FLUS TNs. However, institutional experiences should also be taken under consideration.

Our study has some limitations. It is a retrospective study therefore inaccuracies typical for such a study design were present. The work was performed at a single institution. The number of patients is not high. One of the inclusion criterions of this study was obtaining of the histopathology result, so the study included selection bias, because we evaluated only patients with AUS/FLUS category who underwent surgery. In this retrospective analysis, we could not define a strict and accurate predefined criterion according to which patients were sent to surgery or to a second/third UG-FNAB after the first/second cytological results of AUS / FLUS. The analyzed patients did not have performed molecular tests, so no correlation to AUS/FLUS diagnosis was estimated. We are aware that this information would be the most valuable in cases of an intermediate cytological diagnosis.

The juxtaposition of US features with the number of biopsy repetitions of TNs with consecutive AUS/FLUS diagnoses may simplify the decision-making process in clinical management. Two or three subsequent biopsies with AUS/FLUS diagnosis may increase the risk of malignancy.

## Figures and Tables

**Figure 1 jcm-10-03082-f001:**
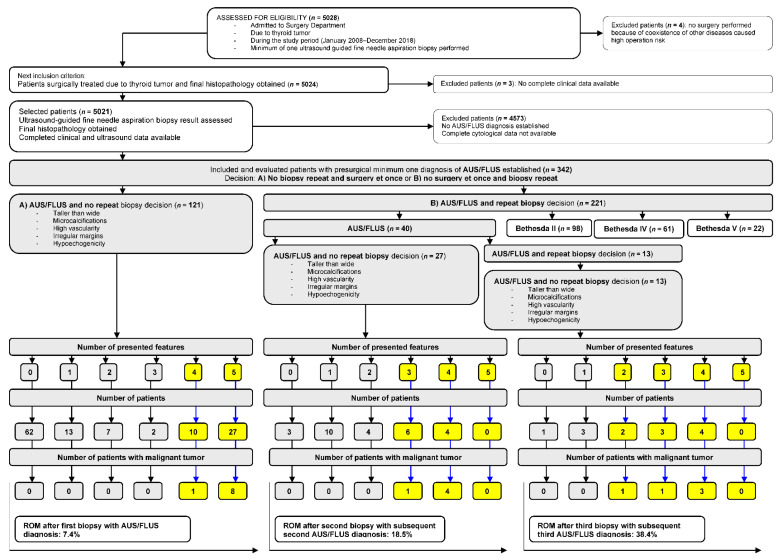
Selection of the study group from 5028 individuals referred for surgery from 2008 to 2018. All participants underwent a minimum of one UG-FNAB with a minimum of one AUS/FLUS diagnosis. All evaluated patients underwent surgery, and histopathology results were obtained in all cases.

**Figure 2 jcm-10-03082-f002:**
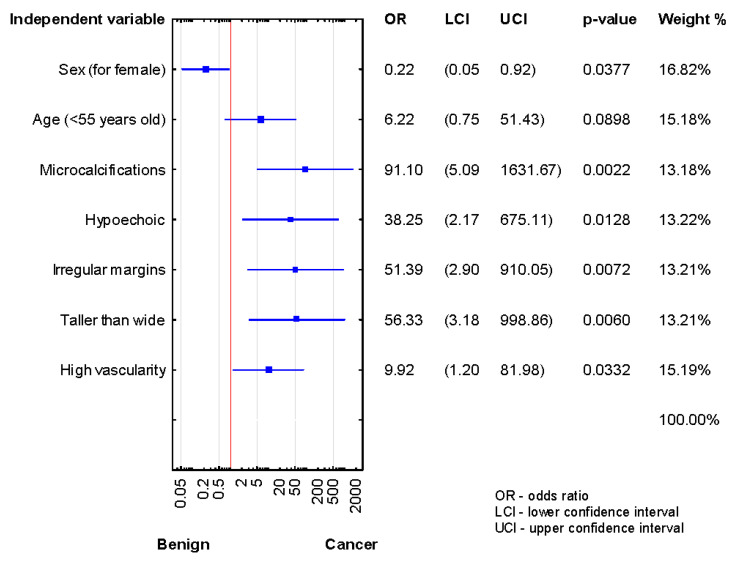
Forrest plot showing the odds ratios for cancer risk in the patients after the first diagnosis of AUS/FLUS (*n* = 121).

**Figure 3 jcm-10-03082-f003:**
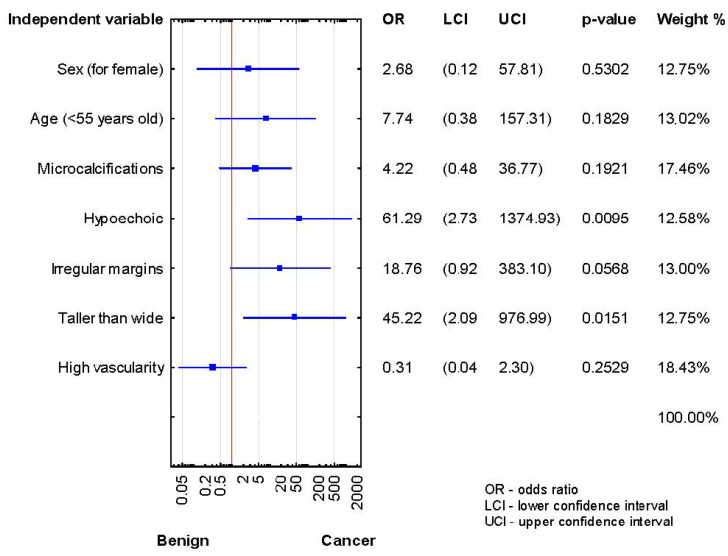
Forrest plot showing the odds ratios for cancer risk in the patients after the second diagnosis of AUS/FLUS (*n* = 27).

**Figure 4 jcm-10-03082-f004:**
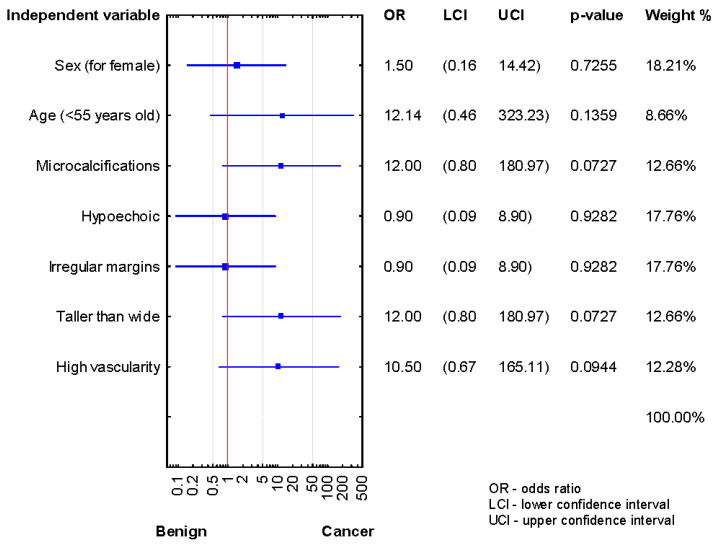
Forrest plot showing the odds ratios for cancer risk in the patients after the third diagnosis of AUS/FLUS (*n* = 13).

**Table 1 jcm-10-03082-t001:** Demographic and clinical characteristics and ultrasound features of 342 patients with AUS/FLUS diagnosis after first UG-FNAB.

Variables		*n* (%) or Mean ± SD
Sex	Female	284 (83.0)
	Male	58 (17.0)
Age (years old)		51.26 ± 15.41
Age	<55 years	191 (55.8)
	>55 years	151 (44.2)
Histopathological diagnosis	Goiter	197 (57.6)
	Adenoma	54 (15.8)
	Thyroiditis	44 (12.9)
	PTC	46 (13.4)
	FTC	1 (0.3)
Nodule size (cm) (median, IQR)		2.00 (1.5–2.6)
	≤1.0 cm	39 (11.4)
	>1.0 ≤ 2.0 cm	139 (40.6)
	>2.0 ≤ 4.0 cm	150 (43.9)
	>4.0 cm	14 (4.1)
**Ultrasound features**		***n* (%)**
Microcalcifications	Yes	72 (21.1)
	No	270 (78.9)
Echogenicity	Hypoechoic	125 (36.5)
	Hyperechoic	217 (63.5)
Irregular margin	Yes	132 (38.6)
	No	210 (61.4)
Taller-than-wide	Yes	111 (32.5)
	No	231 (67.5)
High vascularity	Yes	118 (34.5)
	No	224 (65.5)

AUS/FLUS: atypia of undetermined significance and follicular lesion of undetermined significance; UG-FNAB: ultrasound-guided fine-needle aspiration biopsy; PTC: papillary thyroid cancer; FTC: follicular thyroid cancer.

**Table 2 jcm-10-03082-t002:** Demographic parameters and ultrasound features as predictors for cancer presence in all patients with final diagnosis as AUS/FLUS (*n* = 161). Analysis of contingency tables by Fisher’s exact test and multiple logistic regression analysis (0/1) was used to test the data.

Independent Variables		Benign (*n* = 142)	Cancer (*n* = 19)	*p*-Value (Fisher Exact Test)	OR (+95% CI)	*p*-Value (Wald Test)
		*N* (%)	*N* (%)			
Sex:	Female	117 (82.4)	13 (68.4)	0.210	0.46 (0.15–1.34)	0.154
	Male	25 (17.6)	6 (31.6)			
Age:	<55 years	76 (53.5)	15 (79.0)	0.047 *	3.26 (1.02–10.38)	0.044 *
	>55 years	66 (46.5)	4 (21.0)			
Nodule size:	<2 cm	67 (47.2)	19 (100.0)	<0.0001 *	-	-
	>2 cm	75 (52.8)	0 (0.0)			
Microcalcifications:	Yes	24 (16.9)	15 (79.0)	<0.0001 *	18.43 (5.57–60.98)	<0.0001 *
	No	118 (83.1)	4 (21.0)			
Echogenicity:	Hypoechoic	45 (31.7)	17 (89.5)	<0.0001 *	18.32 (4.01–83.67)	0.0002 *
	Hyperechoic	97 (68.3)	2 (10.5)			
Irregular margins:	Yes	43 (30.3)	17 (89.5)	<0.0001 *	19.56 (4.28–89.46)	0.0001 *
	No	99 (69.7)	2 (10.5)			
Taller-than-wide:	Yes	34 (23.9)	18 (94.7)	<0.0001 *	57.17 (7.24–451.32)	0.0001 *
	No	108 (76.1)	1 (5.3)			
High vascularity:	Yes	66 (46.5)	13 (68.4)	0.089	2.49 (0.89–6.98)	0.079
	No	76 (53.5)	6 (31.6)			

*: statistically significant; AUS/FLUS: atypia of undetermined significance/follicular lesion of undetermined significance.

## Data Availability

The datasets used and/or analyzed during the current study are available from the corresponding author upon reasonable request.

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
