# Peer review of "Atypia and Follicular Lesions of Undetermined Significance in Subsequent Biopsy Result: What Clinicians Need to Know"

_jcm, 2021, doi:10.3390/jcm10143082_

Round 1
Reviewer 1 Report
This is a retrospective study of subjects with thyroid nodule and UG-FNAB showed AUS/FLUS. I have the following comments/suggestions.
- In Materials and Methods, first paragraph, you mentioned analysis of 5,024 medical records but in the third paragraph, n=5,028. Which one is correct?
- Why didn't you use size of thyroid nodule as one of predictive factors for malignancy?
- Why you used age 55 years old cut off, not 60 or 65, etc?
- What was the reason for those patients who underwent surgery? Could this be selective bias as those with more ultrasound features were chosen to have surgery done?
- Could you calculate positive/negative predictive values of ultrasound features (each and in combination) for malignancy? So it will be easier for readers to determine the probability of malignancy in a patient.
Reviewer 2 Report
The authors address a clinical important issue. Indeed, it remains an open question what next step should be taken after an AUS/FLUS cytologic result. The cohort of 342 patients with thyroid nodules with subsequent biopsy results of AUS/FLUS is of considerable size.
Key findings of the paper:
- Significantly higher rates of occurrence of aggressive US features in patients with malignancy (p<0.0001).
- The rate of malignancy after first, second and third biopsies were 24.3%, 50.0%, and 55.5% respectively.
Concerning finding #1: The authors demonstrate 4 ultrasound features (Microcalcification, hypoechogenicity, irregular margin and taler-than-wide shape) to be predictive of malignancy, while hypervascularity is not. This is professionally demonstrated by the authors. However, this is no new knowledge. Indeed, the use of ultrasound features to predict malignancy has been formalized in the widely used and thoroughly tested ACR-TIRADS, EU-TIRADS and other TIRADS systems. The authors fail to address or quote any Thyroid Imaging Reporting and Data System (TIRADS).
In the script no criteria for performing FNA are given. Was FNA performed in every patient that presented to the department with thyroid nodules? The ACR- and EU-TIRADS give specific criteria as when to perform FNA. In the manuscript 62 of 121 patient that had only one FNA presented with no “worrisome” ultrasound features at all. It is very likely that in these patient ACR-TIRADS and EU-TIRADS would not have recommended FNA in the first place.
Furthermore, the authors give no information about nodule size. Nodule size however, is a core criterion in all TIRADS-Systems.
Concerning finding #2: I consider the malignancy rates provided by the authors confusing. The rate of 24.3% applies to patients that had only one FNA but 4 or 5 “worrisome” ultrasound features (9/37). However, the rate in the entire group of patients that had only one FNA is 9 out of 121. A ROM of 50% is quoted for patients that had 2 FNA with AUS/FLUS. However, in this group patients with 3 to 5 “worrisome” ultrasound features are included. In the group of patients who had three FNA with a AUS/FLUS result, patients are considered with 2 to 5 “worrisome” ultrasound features. This is inconsistent.
The distribution of “worrisome” ultrasound features differs between the three groups. Therefore, if the malignancy rates between the groups is to be compared, correction for “worrisome” ultrasound features should be made.
Conclusion:
I would like to suggest a different study design to the authors. First, only nodules in which ACR-TIRADS would have recommended FNA should be included. Then rates of malignancy for nodules with AUS/FLUS cytology should be given stratified by TIRADS-Level (TR3 – TR5). If rates of malignancy differ between the TIRADS-Levels, it could be suggested that patients with AUS/FLUS cytology and high TIRADS-Level should be directed to surgery while patients with a low TIRADS-Level could be followed up or undergo re-FNA.
Material and Methods:
- For descriptive data: Mean and standard deviation should only be used for normally distributed data. Please test data for normal distribution. Non-normally distributed data should be described using median and interquartile range (IQR).
Reviewer 3 Report
Abstract: Significant differences were found between aggressive US features and maligancy in patients after first diagnosis of AUS/FLUS - unclear sentence.
Introduction 8 line before the end : and not to overlook malignancy ?
Materials and methods:
Table 1. The echogenicity is divided into hypoechoic and hyperechoic. There is no isoechoic nodules that are quite often found in that class of TN and is much often found that hyperechoic nodules wchich are most prevalent in the study, that seems very strange???? There is only 1 FTC taht seems also strange????
Figure1 the total number of maligancies is 19.
Results: Table 1 the total number of malignancies is 47 what is different from the information in materials and method.
Based on numbers in materials and methods the statistics can not be performed for 2nd and 3rd UG-FNAB with AUS/FLUS diagnosis.
The data from Fig 1 concerning US features do not support the results of statisctics as only configuration of alltogether 4 or 5 suspicious features can detect malignancy no single us features as presented in the results.
What does it mean: High vascularity?
In the discussion there is some repetition from introduction.
There is a lot of loops in disscusion.
References:
8. Year 2020 not 2021.
Round 2
Reviewer 2 Report
Dear authors,
Thank you for your efforts to adjust your manuscripts according to the reviewer’s recommendations.
However, in my opinion, the authors did not fully address the concerns raised.
- It was suggested that the authors reevaluate the ultrasound studies of the nodules and additionally classify the nodules according to (ACR)-TIRADS. Sadly, the authors failed to do so.
- In the “materials and methods”, section it is not noted who performed the ultrasound readings. From the reply of the authors (“To our analysis we used five ultrasound features, which were obtained from ultrasound descriptions of each thyroid nodules. We did not have any data of evaluation of thyroid nodules according to any TIRADS systems.”) it appears that the authors have no access to the original imaging data but only to written reports. However, ultrasound features are the core subject of the paper. Therefore, in my opinion, it is a prerequisite that access to the ultrasound studies is given and the authors are capable of reevaluating the studies. Do the authors have access to the imaging data? Who performed the ultrasound study and how many years of experience did this person have in performing thyroid ultrasound?
- Nodule size was only added to Table1 but not included into the logistic regression.
- In the previous review I have raised concerns about the ROM stated in table 1 and the abstract: “ Concerning finding #2: I consider the malignancy rates provided by the authors confusing. The rate of 24.3% applies to patients that had only one FNA but 4 or 5 “worrisome” ultrasound features (9/37). However, the rate in the entire group of patients that had only one FNA is 9 out of 121. A ROM of 50% is quoted for patients that had 2 FNA with AUS/FLUS. However, in this group patients with 3 to 5 “worrisome” ultrasound features are included. In the group of patients who had three FNA with a AUS/FLUS result, patients are considered with 2 to 5 “worrisome” ultrasound features. This is inconsistent. The distribution of “worrisome” ultrasound features differs between the three groups. Therefore, if the malignancy rates between the groups is to be compared, correction for “worrisome” ultrasound features should be made.” The authors’ address this concern in the point-to-point reply but failed modify the abstract, table1 or the results section.
Author Response
Journal of July 3, 2021
Clinical Medicine
Dear Editor and Reviewer,
At the very beginning we would like to thank you very much again for the possibility to re-submit our revised manuscript entitled “Atypia and Follicular Lesions of Undetermined Significance in Subsequent Biopsy Result: What Clinicians Need to Know.” Thank you very much for considering it for potential publication in Journal of Clinical Medicine.
We would like to thank you for the very thorough reviews and for the advices and constructive criticism, which have been valuable for improving our paper. All of the suggestions for changes and improvements were very helpful to us, and we have revised the manuscript according to the second recommendations made in the review. All of the changed and deleted portions of the manuscript are marked by using Track Changes Options. According to the reviewer’s instructions we corrected our manuscript point-by-point as follows.
- It was suggested that the authors reevaluate the ultrasound studies of the nodules and additionally classify the nodules according to (ACR)-TIRADS. Sadly, the authors failed to do so.
Dear Editor and Reviewer, the patients included to this study were admitted and treated in one center in years 2008-2018. The data what we can assess comes from medical data base formed on the purpose of this work. There we have clinical, ultrasound and histopathological features. We decided to evaluate the five ultrasound “aggressive” features in patients with AUS/FLUS diagnosis. We do not have the thyroid nodule evaluated according to TIRADS scale. We are very sorry. This is retrospective study, so this is one of the limitations of our work.
- In the “materials and methods”, section it is not noted who performed the ultrasound readings. From the reply of the authors (“To our analysis we used five ultrasound features, which were obtained from ultrasound descriptions of each thyroid nodules. We did not have any data of evaluation of thyroid nodules according to any TIRADS systems.”) it appears that the authors have no access to the original imaging data but only to written reports. However, ultrasound features are the core subject of the paper. Therefore, in my opinion, it is a prerequisite that access to the ultrasound studies is given and the authors are capable of reevaluating the studies. Do the authors have access to the imaging data? Who performed the ultrasound study and how many years of experience did this person have in performing thyroid ultrasound?
All patients admitted and surgically treated in our Center had UG-FNAB performed by two of the authors (KK – general surgeon, oncological surgeon, MD, PhD, and MR – pathologist, MD). All ultrasound examinations were performed by two radiologists with minimum 10 years experience in thyroid ultrasonography. All US features of TNs of every single patient were accurately described and introduced into the medical data base formed for this study. The radiologists were employed in our University Hospital as not scientific workers but they were employed only on service position. All of the patients admitted and surgically treated in our Center had thyroid and neck US performed before surgery. We appreciate their huge work, and we included the acknowledgments in the manuscript: “The authors are grateful to all the staff at the study center who contributed to this work”. We added to the materials and methods section information about who performed US examinations.
- Nodule size was only added to Table1 but not included into the logistic regression.
Data concerning variable “nodule size” were added to Table 2. There were only descriptive data and p-value of Fisher exact test. The probability of nodule size < 2 cm was 100% for patients with TN malignancy and therefore logistic regression analysis could not be performed. Information about this has been added to the Results section.
- In the previous review I have raised concerns about the ROM stated in table 1 and the abstract: “ Concerning finding #2: I consider the malignancy rates provided by the authors confusing. The rate of 24.3% applies to patients that had only one FNA but 4 or 5 “worrisome” ultrasound features (9/37). However, the rate in the entire group of patients that had only one FNA is 9 out of 121. A ROM of 50% is quoted for patients that had 2 FNA with AUS/FLUS. However, in this group patients with 3 to 5 “worrisome” ultrasound features are included. In the group of patients who had three FNA with a AUS/FLUS result, patients are considered with 2 to 5 “worrisome” ultrasound features. This is inconsistent. The distribution of “worrisome” ultrasound features differs between the three groups. Therefore, if the malignancy rates between the groups is to be compared, correction for “worrisome” ultrasound features should be made.” The authors’ address this concern in the point-to-point reply but failed modify the abstract, table1 or the results section.
Dear Editor and Reviewer, thank you very much for this advice and criticism. According to your suggestions we re-drew our flow diagram and entered there new, changed percentages of the patients after first, second, and third UG-FNAB. Indeed, previously we wrote the higher numbers of patients, however it was because we evaluated them according to the “worrisome” features. However we see, that it might be confused for the potential readers. We performed the changes in the abstract, results section and flow diagram
Dear Reviewer,
thank you very much for this detailed, accurate and extremely helpful review.
Thank you.

Reviewer 3 Report
Manuscript significantly imroved.
Author Response
Journal of July 3, 2021
Clinical Medicine
Dear Editor and Reviewer,
At the very beginning we would like to thank you very much again for the possibility to re-submit our revised manuscript entitled “Atypia and Follicular Lesions of Undetermined Significance in Subsequent Biopsy Result: What Clinicians Need to Know.” Thank you very much for considering it for potential publication in Journal of Clinical Medicine.
We performed all suggested corrections and changes, and we can see that all of them were sufficient and acceptable for you. Thank you very much for the statement, that the manuscript is significantly improved.
Thank you again very much.
Kind regards,
Krzysztof Kaliszewski
